# ALKBH5 Stabilized N^6^-Methyladenosine—Modified LOC4191 to Suppress *E. coli*-Induced Apoptosis

**DOI:** 10.3390/cells12222604

**Published:** 2023-11-10

**Authors:** Haojun Xu, Changjie Lin, Chao Wang, Tianrui Zhao, Jinghan Yang, Junhao Zhang, Yanjun Hu, Xue Qi, Xi Chen, Yingyu Chen, Jianguo Chen, Aizhen Guo, Changmin Hu

**Affiliations:** 1Department of Clinical Veterinary Medicine, College of Veterinary Medicine, Huazhong Agricultural University, Wuhan 430070, China; xuhaojun@webmail.hzau.edu.cn (H.X.); 1875155626@outlook.com (C.L.); ztroct@163.com (T.Z.); yangjinghan@link.cuhk.edu.hk (J.Y.); zhangjunhao@webmail.hzau.edu.cn (J.Z.); 13618639606@163.com (Y.H.); 15776560206@163.com (X.Q.); 2State Key Laboratory of Agricultural Microbiology, Huazhong Agricultural University, Wuhan 430070, China; wangchao666@webmail.hzau.edu.cn (C.W.); chenxi@mail.hzau.edu.cn (X.C.); chenyingyu@mail.hzau.edu.cn (Y.C.); chenjg@mail.hzau.edu.cn (J.C.); aizhen@mail.hzau.edu.cn (A.G.); 3Department of Preventive Veterinary Medicine, College of Veterinary Medicine, Huazhong Agricultural University, Wuhan 430070, China; 4Hubei Hongshan Laboratory, Huazhong Agricultural University, Wuhan 430070, China

**Keywords:** N^6^-methyladenosine, long non-coding RNA, apoptosis, bovine mammary epithelial cells, *E. coli*

## Abstract

*E. coli* is a ubiquitous pathogen that is responsible for over one million fatalities worldwide on an annual basis. In animals, *E. coli* can cause a variety of diseases, including mastitis in dairy cattle, which represents a potential public health hazard. However, the pathophysiology of *E. coli* remains unclear. We found that *E. coli* could induce global upregulation of m^6^A methylation and cause serious apoptosis in bovine mammary epithelial cells (MAC-T cells). Furthermore, numerous m^6^A-modified lncRNAs were identified through MeRIP-seq. Interestingly, we found that the expression of LOC4191 with hypomethylation increased in MAC-T cells upon *E. coli*-induced apoptosis. Knocking down LOC4191 promoted *E. coli*-induced apoptosis and ROS levels through the caspase 3–PARP pathway. Meanwhile, knocking down ALKBH5 resulted in the promotion of apoptosis through upregulated ROS and arrested the cell cycle in MAC-T cells. ALKBH5 silencing accelerated LOC4191 decay by upregulating its m^6^A modification level, and the process was recognized by hnRNP A1. Therefore, this indicates that ALKBH5 stabilizes m^6^A-modified LOC4191 to suppress *E. coli*-induced apoptosis. This report discusses an initial investigation into the mechanism of m^6^A-modified lncRNA in cells under *E. coli*-induced apoptosis and provides novel insights into infectious diseases.

## 1. Introduction

*E. coli* is a Gram-negative bacterium of the family Enterobacteriaceae. It has been primarily isolated from the intestinal contents of various vertebrates, predominantly mammals, and is also present in the environment [1]. Pathogenic *E. coli* is responsible for over 160 million cases of dysentery and one million fatalities worldwide on an annual basis [2]. Moreover, *E. coli* has been identified as a prevalent pathogen responsible for mastitis in dairy cows, resulting in decreased production and quality of milk. Consumption of contaminated milk poses a significant threat to human health due to the potential transmission of resistant pathogens, resulting in serious public health hazards [3]. Although several other pathogens were also isolated from bovines with mastitis [4], including *Streptococcus* and *Staphylococcus aureus*, these bacteria usually only cause mild mastitis. However, 50.6% of bovines with severe mastitis are caused by *E. coli*, which leads to extremely high mortality within 2 weeks. Consequently, exploring the pathophysiology of *E. coli* is imperative.

Several studies have verified that *E. coli* infection results in mitochondrial damage and a subsequent increase in reactive oxygen species (ROS) production [5], which finally promotes apoptosis [5,6,7]. Pathogens cause differential expression of apoptosis-related transcripts, particularly some long noncoding RNAs (lncRNAs) [8,9,10]. It is understood that the intricate regulatory mechanism of lncRNAs is crucial for the precise expression of genes, as opposed to the previously held belief that they were merely transcriptional noise. Recent research has demonstrated that lncRNAs are extensively involved in various biological processes, including inflammation [11], the cell cycle [12], and apoptosis [13].

N^6^-methyladenosine (m^6^A) is a prevalent epigenetic modification found in lncRNAs, with a widespread distribution across all eukaryotic cells. It is known to regulate RNA splicing [14], stability [15], and decay [16]. Plenty of research has shown that m^6^A modification is closely related to infectious diseases [17,18,19]. Furthermore, there is a great deal of research demonstrating that AlkB Homolog 5 (ALKBH5) is a mammalian RNA demethylase, and it may play an important role in regulating apoptosis through an association with m^6^A-modified lncRNA. Recently, a study showed that ALKBH5 could reduce lncRNA TP53TG1 stability and downregulate its expression, ultimately promoting apoptosis in gastric cancer cells [20]. Another study demonstrated that ALKBH5-mediated m^6^A demethylation stabilized the lncRNA CASC8 transcript, resulting in lncRNA CASC8 upregulation. Moreover, lncRNA CASC8 was verified to promote cancer cell apoptosis through the BCL2/caspase 3 pathway [21]. Hence, lncRNAs exhibiting m^6^A modification have been deemed a potential target in apoptosis [22,23]. However, there has been no investigation into the relationship between m^6^A-modified lncRNAs and *E. coli* infection.

In order to ascertain the involvement of lncRNAs with m^6^A modifications in apoptosis, MeRIP-seq was used to predict and characterize the function of lncRNAs in MAC-T cell apoptosis induced by *E. coli*. Our investigation identified LOC4191 as a crucial player in the *E. coli*-induced apoptosis of MAC-T cells. ALKBH5 was found to stabilize m^6^A-modified LOC4191 to suppress *E. coli*-induced apoptosis through the caspase 3–PARP pathway. Our study thus elucidates the m^6^A-modified mechanism of apoptosis in MAC-T cells, thereby presenting a potential target for the diagnosis and treatment of *E. coli* infection.

## 2. Methods

### 2.1. Cell Lines and Bacteria

Professor Mark Hanigan of Virginia Tech University donated MAC-T cells (immortalized bovine mammary epithelial cells). The MAC-T cells were cultured by adapting our previous method [24,25].

Professor Wang Xiangru of Huazhong Agricultural University donated *E. coli* (ATCC 25922). The *E. coli* was cultured by adapting our previous method [24,25].

### 2.2. Inactivated E. coli-Induced Apoptosis in MAC-T Cells

*E. coli* was inactivated by a 63 °C water bath for 30 min, followed by plate coating to ensure bacterial inactivation. Then, 2 × 10^5^ cells were added to each well in a 6—well plate (Corning, Somerville, MA, USA) after cell counting. Cells attached to the wall after 12 h. A total of 2 × 10^6^ inactivated *E. coli* was added to each well (MOI = 10), and 24 h coculture was applied in MAC-T cells when inducing apoptosis.

### 2.3. siRNA Transfection

Cells were added to a 6-well plate and cultured at 37 °C, 5% CO_2_. Then, 2 L of jetPRIME (Polyplus, Strasbourg, France) was incubated with 50 nM siRNA for 10 min. The mixture was added to the cells for 24 h coculture. After RNA extraction, the efficiency of transfection was calculated. The sequences of siRNAs are as follows: si-ALKBH5-1 (5′—GCUGCAAGUUCCAGUUCAATT—3′); si-ALKBH5-2 (5′—GCGCCGUCAUCAACGACUATT—3′); siLOC4191-1 (5′—GGCCGAUAGGAUGGGAAUUTT—3′); si-LOC4191-2 (GCAGUCGACAUUGCUGGCATT—3′).

### 2.4. RNA Extraction and RT-qPCR

Cold PBS (Hyclone, Tauranga, New Zealand) was used to wash the cells 3 times, and 1 mL Trizol (Invitrogen, Carlsbad, CA, USA) was added to each well. The cells were lysed and collected in EP tubes. Then, 200 μL chloroform was added, followed by a 30 s vortex and centrifugation at 12,000 rpm at 4 °C for 10 min. Then, 500 μL of the supernatant was collected in a new EP tube. Next, 500 μL of isopropanol was added and mixed upside down softly. The mixture was allowed to stand for 10 min at 4 °C, followed by centrifugation at 12,000 rpm at 4 °C for 15 min. The supernatant was removed, and the RNA pellet could be visible. Then, 1 mL 75% ethanol was added to wash the RNA, followed by centrifugation at 7500 rpm at 4 °C for 5 min. The supernatant was removed, and RNA was dried for 15 min. Next, 20 μL DEPC water was added, then incubated at 58 °C in a water bath for 10 min. The purified RNA was obtained.

As a quality control index for RNA purity, OD260/OD280 values between 1.8 and 2.0 were determined using the NanoDrop 2000 instrument (Thermo Fischer Scientific, Waltham, MA, USA). We used denaturing agarose gel electrophoresis to measure RNA integrity and contamination with gDNA. The sample was stored at −80 °C for later analysis.

Reverse transcription of the RNA samples was performed using HiScript III RT SuperMix for qPCR (+gDNA wiper) (Vazyme, Nanjing, China); 4 μL of 4× gDNA wiper Mix was added into 1 μg RNA and RNAase—free ddH_2_O was added to 16 μL. The mixture was incubated at 42 °C for 2 min to remove genomic DNA. Reverse transcription was performed by adding 4 μL of 5× HiScript III qRT SuperMix. The mixture was incubated at 37 °C for 15 min and 85 °C for 5 s to obtain cDNA.

The expression of cDNA in different groups of samples was detected using AceQ qPCR SYBR Green Master Mix (Vazyme, China) in a ViiA7 Real-time PCR machine (Applied Biosystems Inc., Foster City, CA, USA). The final volume of the real-time PCR reaction is 20 μL, including 10 μL of 2× AceQ qPCR SYBR Green Master Mix, 0.4 μL of upstream primer (10 μM), 0.4 μL of downstream primer (10 μM), 0.4 μL of 50× ROX Reference Dye 2, 3 μL of cDNA template, and 5.8 μL of ddH_2_O. The reaction was completed as follows: 95 °C for 5 min (1 cycle), 95 °C for 10 s (40 cycles), 60 °C for 30 s (40 cycles). Appendix A shows the PCR primer sequences.

### 2.5. Flow Cytometry

Apoptosis detection: Cold PBS was used to wash the cell samples 3 times. Cells were digested by trypsin, and the digestion was terminated with DME/F-12 medium containing 10% FBS. Cells were collected by centrifugation at 1500 rpm for 5 min. The samples were washed 3 times in PBS by centrifugation at 1500 rpm for 5 min. The supernatant was discarded. Annexin V-FITC Apoptosis Detection Kit (Beyotime, Shanghai, China) was used to detect apoptosis. Then, 100 μL of binding buffer was added to resuspend the cells. MAC-T cells were stained with Annexin V-FITC and Propidium Iodide (PI) for 10 min before being detected by CytoFLEX-LX (Becton Dickinson, Franklin Lakes, NJ, USA).

ROS detection: ROS Assay Kit (Beyotime, China) was used. DME/F-12 medium without FBS was used to dilute DCFH-DA (final concentration equal to 10 μM). After collecting the digested cells, medium without FBS was used to wash the cells 2 times. Cells were collected by centrifugation at 1500 rpm for 5 min, followed by discarding the supernatant. Cells were resuspended and incubated in the diluted DCFH-DA at 37 °C for 20 min. Medium without FBS was used to wash the cells 3 times before detection.

Cell cycle detection: The cells were washed with cold PBS, followed by centrifugation at 1500 rpm for 5 min, and the supernatant was discarded. Then, 70% ethanol was used to fix the cells for 12 h. The cells were centrifuged at 1500 rpm for 5 min, and the supernatant was discarded. Cold PBS was used to wash the cells once. A Cell Cycle and Apoptosis Analysis Kit (Beyotime, China) was used. The staining mixture is 535 μL total, including 500 μL of staining buffer, 25 μL of PI dye, and 10 μL of RNase A. The samples were incubated with the staining mixture at 37 °C for 30 min before detection.

### 2.6. Transmission Electron Microscopy

First, 2.5% glutaraldehyde was used to pre-fix the cell samples. Dehydration was performed with an ascending ethanol series: 15 min in 50% ethanol, 8 h in 70% ethanol, 25 min in 90% ethanol, and 5 min in 100% ethanol. Samples were then incubated in pure resin for 6 h, placed in embedding molds, and incubated at 70 °C for 60 h. The embedded samples were used for TEM slide preparation. The transmission electron microscope H7650 (HITACHI, Tokyo, Japan) was used to observe the slide.

### 2.7. Western Blot

RIPA reagent (Sigma, St. Louis, MO, USA) containing protease inhibitors and phosphatase inhibitors (Roche, Shanghai, China) was added to the cell samples, and the cells were lysed for 30 min on ice. The supernatant was collected by centrifugation at 12,000 rpm, 4 °C for 10 min. A BCA kit (Beyotime, China) was used to detect the protein concentrations of each sample, and proteins were adjusted to equal concentrations. The protein samples were mixed with 5× protein loading buffer and boiled for 10 min. Proteins were separated using SDS-PAGE on 10% polyacrylamide gels at 100 V for 90 min and then were transferred to polyvinylidene difluoride (PVDF) membranes (Millipore, Burlington, MA, USA) at 100 V for 70 min. The PVDF membrane was put into 5% skimmed milk and sealed for 4 h. The membrane was washed 3 times with Tris-buffered saline with 0.15% Tween-20 (TBST) for 5 min. After that, the membrane was incubated with the primary antibody at 4 °C for 12 h and the secondary antibody at room temperature for 1 h.

The following primary antibodies were used: cleaved caspase 3 (Proteintech, Shanghai, China), PARP (Proteintech, China), β-actin (Bioss, Beijing, China), and MAPK-related antibody (CST, Danvers, MA, USA). The HRP secondary antibody was used (Invitrogen, USA).

### 2.8. FISH

After cell counting, 5 × 10^4^ cells were seeded in 24-well culture plates per well plated with cell-climbing slices. After cell adhesion, cells were washed with PBS 3 times and fixed by 4% paraformaldehyde for 10 min and permeabilized by 0.5% Triton X-100 for 5 min. The fixed samples were washed with PBS 3 times. A Ribo^TM^ Fluorescent In Situ Hybridization Kit (Ribobio, Guangzhou, China) was used. First, 200 μL of pre-hybridization buffer was added to the plates and incubated at 37 °C for 30 min. Then, 100 μL of hybridization buffer (containing 2.5 μL LOC4191 FISH Probe Mix binding with Cy3) was added to the plate and incubated at 37 °C for 12 h. Saline sodium citrate buffer (SSC) was used to wash the samples 3 times at 42 °C for 5 min. PBS was used to wash the sample once, followed by staining the nucleus with DAPI for 10 min. A confocal laser scanning microscope was used to detect the fluorescent signal.

### 2.9. RNA Pull Down

The interaction between LOC4191 and binding protein was examined using the Pierce^TM^ Magnetic RNA-Protein Pull-Down Kit (Thermo Fisher Scientific, USA) following the instructions of the manufacturer. After collecting 50 μL of cell precipitation, 300 μL of IP lysate was added and vortexed for 10 s, and the samples were incubated on ice for 30 min. The supernatant was collected after centrifugation at 12,000 rpm, 4 °C for 10 min, and 50 μL of the total protein was taken as input. The sense tube and antisense tube (1.5 mL) were added with 50 μL of streptavidin magnetic beads and put on a magnetic stand for 1 min to discard the supernatant. The tubes were washed with 500 μL of buffer A twice, 500 μL of buffer B once, and 20 mM Tris (pH7.5) twice. After washing, 500 μL of 1× RNA capture buffer and 50 pmol biotinylated RNA probes (5′—AUCAGCCAGCACCGACUUGCC—3′) were added to the tube. The mixture was incubated at room temperature for 30 min. RNA probes were bound to streptavidin magnetic beads. The sense tube and antisense tube were put on a magnetic stand for 1 min to collect the beads. The tubes were washed with 20 mM Tris (pH 7.5) twice, followed by adding 20 μL of 10× protein-RNA binding buffer, 60 μL of 50% glycerol, and 2 mg of total protein. The mixture was incubated at 4 °C for 1 h. Then, the beads-probe-protein complex was obtained. The tubes were put on a magnetic stand for 1 min to collect the beads. The tube was washed with 200 μL of 1× washing buffer twice, and the protein was eluted with 100 μL elution buffer for 15 min. The supernatant was collected and restored at −80 °C. A total of 3 groups of protein was obtained, including protein pulled down by LOC4191 sense probe (probe group), protein pulled down by LOC4191 antisense probe (control group), and total protein (input group). The RNA-binding protein was analyzed by Mass Spectrometry (MS) (5600-plus, AB SCIEX, Framingham, MA, USA) and Western Blot (WB).

### 2.10. Molecular Docking Analysis

Autodock Vina 1.2.2 was employed. N^6^-methyladenosine molecular structure was obtained from PubChem Compounds (CID, 102157). The 3D structure of HNRNPA1 (PDB ID, 1HA1; resolution, 1.5 Å) was obtained from the PDB website. We converted all protein and molecular files into PDBQT format, excluding all water molecules and adding polar hydrogen atoms for docking analysis. The grid box was centered to cover each protein’s domain. A 0.05 nm grid point distance was selected for the grid box, which was set to 30 Å × 30 Å × 30 Å.

### 2.11. M^6^A-Modified lncRNA Library Construction

The MeRIP-lncRNA sequencing was carried out at Cloud-Seq Biotech (Shanghai, China). A total of 6 samples were sent for sequencing under the following procedure: (1) To enrich lncRNA, Cloud-Seq’s lncRNA enrichment kit was used. (2) A sequencing library was constructed using the Library Kit (NEB, Ipswich, MA, USA). (3) BioAnalyzer 2100 (Agilent Technologies, Santa Clara, CA, USA) was used to control and quantify the library, and Illumina was used to perform sequencing.

Through Cutadapt and Bowtie2, raw reads obtained from Illumina sequencing were purified and got good quality control. We aligned reads to the bovine reference genome (ARS-UCD1.2) through Hisat2 software (2.2.0). The raw sequencing data contains 3 control cell samples (C1, C2, C3) and 3 *E. coli*-induced cell samples (E1, E2, E3). The overlap of methylation peaks on lncRNAs in multiple samples was calculated by Bedtools [26], and the identified peaks were obtained. The differential m^6^A peaks were compared between the control group and the *E. coli* group by diffReps [27] (*p*-value < 0.0001, Fold change > 2).

Differential m^6^A modification levels within the lncRNAs were considered. DAVID software [28] was used to perform GO and KEGG enrichment analysis. R language was performed in other bioinformatic analyses.

Flowjo_V10 and Prism 8 were widely used in this study. Results from three independent experiments are shown as means (±SEMs). A *p*-value < 0.05 was considered statistically significant when the Student’s *t*-test differentiated groups.

## 3. Results

### 3.1. M^6^A-Modified lncRNAs in Apoptosis of MAC-T Cells Induced by E. coli

MeRIP-seq was used to investigate the role of lncRNAs with m^6^A modification in MAC-T cell apoptosis induced by *E. coli*. With rigorous quality control, 99.122–99.993% of net reads were obtained (Appendix A).

Subsequent analysis of the MeRIP-seq data (Appendix A) revealed the presence of 714 m^6^A peaks in 649 lncRNAs in the control group and 345 peaks in 324 lncRNAs in the *E. coli* group (Appendix A). Furthermore, 286 peaks were identified in 276 lncRNAs in both groups. While the majority of lncRNAs were found to be modified with a single m^6^A peak, multiple m^6^A peaks in a single lncRNA were also observed (Appendix A). Furthermore, the analysis of m^6^A peak distribution indicated the predominance of chromosome 3 in the control group and chromosome 10 in the *E. coli* group (Appendix A). Additionally, the length and origin of m^6^A-modified lncRNAs exhibited similar patterns in both groups, with the majority of lncRNAs being less than 5000 bp in length (Appendix A). Intergenic m^6^A-modified lncRNAs were found to be the most abundant (Appendix A).

### 3.2. Differential m^6^A-Modified lncRNAs in Apoptosis of MAC-T Cells Induced by E. coli

To explore the function of m^6^A-modified lncRNAs in *E. coli* induction, we identified a total of 920 differential m^6^A methylation peaks in 288 lncRNAs, with 330 hypermethylated peaks in 119 lncRNAs (e.g., LOC112443802) and 590 hypomethylated peaks in 169 lncRNAs (e.g., LOC101903744) (Appendix A). The differential m^6^A peaks between the control and *E. coli* groups were illustrated through data visualization analysis conducted by IGV (Appendix A). The chromosome distribution was further elucidated, revealing that hypermethylated lncRNAs were predominantly enriched on chromosome 19, while hypomethylated lncRNAs were primarily enriched on chromosome 17 (Appendix A). Additionally, altered lncRNAs with hypermethylation exhibited significantly shorter lengths compared to those with hypomethylation (Appendix A). Almost half of the lncRNAs with hypermethylation originated from intron sense-overlapping genes. Conversely, lncRNAs with hypomethylation from all sources displayed an average pattern, with an increase in the intergenic region in the hypermethylation section (Appendix A).

To elucidate the potential functions of different m^6^A-modified lncRNAs in MAC-T cell apoptosis induced by *E. coli*, GO (Appendix A) and KEGG (Appendix A) analyses were conducted with a *p*-value threshold of 0.05. The GO annotation revealed that the m^6^A-modified lncRNAs were primarily enriched in the cell cycle and immune response (Appendix A). The KEGG enrichment analysis revealed that they were linked to inflammation- and apoptosis-related pathways (Appendix A). This indicates a close relationship between apoptosis and these altered m^6^A-modified lncRNAs.

### 3.3. E. coli-Induced Apoptosis and Involvement of m^6^A in MAC-T Cells

To examine the potential for *E. coli* to induce apoptosis in MAC-T cells, we used heat-inactivated *E. coli* as a stimulus for a 24-h period. Our findings indicate that the expression of IL-1β, IL-6, and TNF-α was significantly elevated in MAC-T cells induced by *E. coli* (Figure 1A–C). Additionally, our results demonstrate that apoptosis rate (Figure 1D) and ROS levels (Figure 1E) in MAC-T cells were increased by *E. coli* induction. We then used transmission electron microscopy to provide a more visual representation of *E. coli*-induced apoptosis in MAC-T cells (Figure 1F–H). Remarkably, it revealed the presence of acute cytosolic voids, the disappearance of the nuclear membrane, and chromatin dissolution in MAC-T cells with apoptosis induced by *E. coli* (Figure 1G,H). Immunoblotting of lysates revealed an increase in phosphorylated p38 and p65 proteins, while phosphorylated JNK and ERK showed no significant changes in MAC-T cells with apoptosis induced by *E. coli* (Figure 1I,J). The *E. coli* group exhibited distinct patterns in the MAC-T cell cycle, with more cells arrested in the S and G2 phases (Figure 1K,L).

Interestingly, we found that m^6^A modification was involved in apoptosis. The results indicate significant upregulation of global m^6^A modification (Figure 1M). Additionally, the expression of methylase METTL3, METTL14, and WTAP increased (Figure 1N–P). Conversely, the expression of demethylase ALKBH5, exhibited significant upregulation, while FTO showed no significant changes (Figure 1Q,R).

### 3.4. ALKBH5 Silencing Promoted E. coli-Induced Apoptosis in MAC-T Cells

Even though global m^6^A modification showed upregulation, the expression of eraser ALKBH5 was observed to be significantly high in MAC-T cells with apoptosis induced by *E. coli* (Figure 2A). In order to explore its role, we successfully knocked down ALKBH5 (Figure 2B,C). Interestingly, we observed an increase in cleaved caspase 3 and a decrease in PARP (Figure 2D). Furthermore, it resulted in increased apoptosis and elevated ROS levels (Figure 2F). Additionally, we observed a rise in phosphorylated proteins JNK, p38, ERK, and p65 in MAC-T cells with apoptosis induced by *E. coli* (Figure 2G,H).

### 3.5. Characteristics of M^6^A-Modified LOC4191 in MAC-T Cells

Additionally, a conjoint analysis of the MeRIP-seq data and the previously published lncRNA-seq data [29] was conducted to demonstrate the relationship between m^6^A modification and lncRNA expression (Figure 3A). In screening targets with crucial roles in *E. coli*-induced apoptosis, 20 lncRNAs attracted our attention, and we used RT-qPCR to confirm that their expression changed in the *E. coli* group (Appendix A). We found that the expression of LOC4191 was steadily upregulated in MAC-T cells with *E. coli*-induced apoptosis (Figure 3B). To validate the precise location of the m^6^A site, we performed MeRIP-qPCR and confirmed that the m^6^A modification of LOC4191 decreased in the *E. coli* group (Figure 3C). The hypomethylation in LOC4191 was illustrated by IGV software (version 1.13.11) (Figure 3D), with the m^6^A site identified at No. 635 adenylate (Figure 3E). The nuclear localization of LOC4191 was demonstrated by fluorescence in situ hybridization (FISH) (Figure 3F) and further confirmed by a cytoplasm/nucleus isolation experiment (Figure 3G). Additionally, the coding potential of LOC4191 was predicted to be similar to that of lncRNA XIST, rather than actin, based on a CPC2 database analysis (Figure 3H).

### 3.6. M^6^A-Modified LOC4191 Silencing Promoted Apoptosis in MAC-T Cells

As previously mentioned, the increased expression of LOC4191 with hypomethylation may have an impact on MAC-T cell apoptosis induced by *E. coli*. Further investigation into the role of LOC4191 is imperative. Upon silencing of LOC4191, an elevation in ROS levels (Figure 4A) and an increase in apoptosis rate (Figure 4B) were observed in *E. coli*-induced MAC-T cells. Moreover, the expression of cleaved caspase 3 increased, and PARP decreased over the same period (Figure 4C).

### 3.7. M^6^A-Modified LOC4191 Is the Target of ALKBH5

In order to confirm the relationship between m^6^A-modified LOC4191 and ALKBH5, we knocked down ALKBH5 and observed a significant increase in the m^6^A modification level of LOC4191 (Figure 4D), resulting in decreased expression (Figure 4E). Then, we employed actinomycin D to impede cellular transcription and assessed the expression level of LOC4191 at six time points (0, 1, 2, 4, 8, and 12 h). Our findings revealed a noteworthy acceleration in the degradation rate of LOC4191 in the ALKBH5 knockdown group compared to the control group (Figure 4F).

### 3.8. M^6^A-Modified LOC4191 Is Recognized by hnRNP A1 Reading Protein

Using RNA pulldown and mass spectrometry techniques, a total of 351 associated proteins were identified in the probe group (Appendix A), and the top 10 proteins pulled down by LOC4191 were closely related to apoptosis (Table 1). Additionally, an analysis of COG2 (Appendix A), IPR (Appendix A), subcellular localization (Appendix A), GO (Appendix A), and KEGG (Appendix A) was conducted to predict the function of the proteins pulled down by LOC4191. According to the bioinformatics predictions, a strong correlation was observed between the LOC4191 pulldown proteins and immune response.

To comprehensively explore the role of m^6^A, the interaction between the pulled-down hnRNP A1 protein and LOC4191 was confirmed via Western blot (Figure 5A). To assess the affinity of hnRNP A1 for its m^6^A targets, molecular docking analysis was conducted, resulting in the identification of binding poses and interactions (Figure 5B). The findings indicate that hnRNP A1 engages with its m^6^A targets via discernible hydrogen bonds and robust electrostatic interactions, with a binding energy of -5.564 kcal/mol, indicating remarkably stable binding. These results suggest that hnRNP A1 may recognize LOC4191 with m^6^A modification.

## 4. Discussion

Pathogenic *E. coli* is a globally prevalent bacterium that has caused millions of fatalities. It is also a frequently encountered pathogen in agricultural animals. For example, *E. coli* is the primary pathogen of acute mastitis in bovines [30]. Contamination of milk by *E. coli* could lead to serious public health problems and greatly endanger human lives [31,32]. In pathogenesis, *E. coli* is a typical Gram-negative bacterium, and its membrane contains various kinds of antigens, including LPS, that could induce a serious immune response in the host, while other Gram-positive bacteria mainly secrete exotoxins. Overall, it is necessary to explore the pathogenesis of *E. coli*-induced apoptosis.

Given that MAC-T cells serve as the primary barrier against bacterial infection, investigating the pathophysiology of *E. coli* is of paramount importance. Numerous studies have reported that m^6^A modifications are detected in various noncoding RNAs, including lncRNA, circRNA, and tRNA [33,34]. Notably, some investigations have highlighted the crucial role of m^6^A-modified lncRNAs in regulating apoptosis [35,36,37,38]. In our study, we detected a substantial quantity of novel m^6^A peaks through MeRIP-sequencing. Given the uncertain role of m^6^A-modified lncRNAs, KEGG and GO analyses were conducted to determine the functional enrichment of the altered genes, which was found to be closely related to inflammatory responses and apoptosis [39,40].

In our study, IL-1β, IL-6, and TNF-α were upregulated, and flow cytometry showed greater apoptosis rates and ROS levels when MAC-T cells were induced by *E. coli.* IL-1β, IL-6, and TNF-α have been documented as important cytokines in apoptosis caused by bacteria [41,42,43,44,45,46,47,48]. Moreover, the presence of acute cytosolic voids, the disappearance of the nuclear membrane, and the chromatin dissolution observed by transmission electron microscopy are directly indicative of apoptosis [49,50], which was also observed in our study. Furthermore, apoptosis is a crucial mechanism in programmed cell death, often concomitant with ROS upregulation [51,52,53,54] and cell cycle disturbances [55,56,57]. Interestingly, we found that the amounts of phosphorylated proteins p38 and p65 increased in MAC-T cells with apoptosis induced by *E. coli*. Numerous studies have demonstrated that the MAPK pathway is a classical pathway for apoptosis, with broad regulatory effects on various infectious diseases [48,58,59,60]. Furthermore, the MAPK pathway has been widely confirmed to collaborate with NF-κB [61,62,63]. P65 is a crucial nuclear transcription factor [60,64,65,66,67], and p38 has the ability to activate p65 and trigger apoptosis [60,68,69].

Interestingly, we found that *E. coli*-induced apoptosis caused global upregulation of m^6^A modification in MAC-T cells. Nevertheless, there remains a significant debate regarding the essential molecules that contribute more significantly to apoptosis, namely, writer proteins encoded by METTL3, METTL14, and WTAP, the eraser protein encoded by ALKBH5, and FTO. Several studies have suggested that the ALKBH5 gene may have potential for antimicrobial purposes [70,71]. Presently, many research endeavors are concentrated on investigating the impact of ALKBH5 on tumor apoptosis [72,73], a crucial area of inquiry. However, it is equally imperative to explore the effects of ALKBH5 on apoptosis induced by bacteria such as *E. coli*. In our study, it was observed that global m^6^A modification was upregulated, along with demethylase ALKBH5 expression. To further confirm the function of ALKBH5 in *E. coli*-induced apoptosis, we knocked down ALKBH5 by two siRNAs. Knockdown of ALKBH5 promoted apoptosis when MAC-T cells were induced by *E. coli*. Moreover, we found that the knockdown efficiency of siALKBH5-2 is less than that of siALKBH5-1, but its impact on apoptosis is stronger. The remaining function of ALKBH5 seemed not positively correlated with its amount. To solve this, further experiments are needed to knock out ALKBH5 completely by the Crispr–Cas9 system. Overall, we proved that ALKBH5 plays an important role in regulating *E. coli*-induced apoptosis.

By screening all of the m^6^A-modified lncRNAs detected in MeRIP-sequencing, we found that m^6^A-modified LOC4191 plays a significant role in apoptosis. FISH is one of the most common methods to identify certain RNA subcellular localization [74], which is the basis for further investigation. Moreover, many sequencing results have illustrated that only a small fraction of the m^6^A consensus motifs in lncRNAs are modified [75,76,77]. Therefore, m^6^A sites need to be accurately verified [78]. In our study, we first identified the nuclear localization of LOC4191. Then, we conducted MeRIP-qPCR to validate the specific m^6^A site of LOC4191. Finally, we found that the amount of LOC4191 with hypomethylation increased and, therefore, regulated apoptosis when MAC-T cells were induced by *E. coli*.

Consequently, we aimed to identify whether LOC4191 was regulated by demethylase ALKBH5 in MAC-T cells with apoptosis induced by *E. coli*. Notably, previous research has demonstrated that m^6^A modification primarily regulates the stability of target genes [79]. Researchers have found that loss of ALKBH5 is associated with poorer survival in patients with cancer [80]. Another study investigated the role of ALKBH5 in cancer development, indicating that it may stabilize oncogenes and accelerate cancer progression [81]. In our study, the knockdown of ALKBH5 decreased the expression of LOC4191 with hypermethylation by reducing its stability. Therefore, we successfully confirmed that m^6^A-modified LOC4191 is a target of ALKBH5.

Proteins serve as the primary agents of biological functions, and RNA-binding proteins, which are ubiquitous in organisms, extensively regulate physiological processes [82,83,84]. In our study, Mass spectrometry was used to identify many binding proteins pulled down by LOC4191, which mainly exhibit functions associated with bacterial infection, immune response, and apoptosis. The top 10 pulled-down proteins in Table 1 provide further direction in studying *E. coli*-induced cell injury, such as HSP, Acetyl-CoA carboxylase, and Annexin A2.

Since the precise mechanism of m^6^A reading proteins remains uncertain, it is currently established that nuclear readers mainly comprise the hnRNP protein family, while cytoplasmic readers comprise the IGF2BP and YTH protein families [79,85]. In order to examine the m^6^A reading protein responsible for the degeneration of LOC4191, we conducted RNA pull-down, molecular docking analysis, and Western blot. We successfully screened a reading protein hnRNP A1. The hnRNP family could regulate different biological processes, and they will most likely interact with various RNAs.

Convincingly, hnRNP A1 is a reading protein localized to the nucleus, and this is consistent with the subcellular localization of LOC4191 in the nucleus. Moreover, hnRNP A1 has been verified that it could decrease the stability of its target gene in an m^6^A-dependent manner [86]. In our study, we demonstrated an increase in the m^6^A modification level of LOC4191 after knocking down demethylase ALKBH5. This promotes the binding between m^6^A-modified LOC4191 and hnRNP A1, causing a decrease in LOC4191 stability.

However, our study still had some limitations. Even though heat-inactivated *E. coli* has all the same immunogenicity, future studies will need to use live pathogens, which would better show the host–pathogen interaction and reflect the actual clinical problem. Even though we revealed that m^6^A affects the binding between hnRNP A1 and LOC4191, we would like to try knocking out the single methylated adenylate within the lncRNAs in the future and verify whether this could prevent the binding and slow down the degradation of the target genes. Additional investigation is required to fully elucidate the role of hnRNP A1 as an m^6^A-reading protein, as well as expand upon the limited literature available on the function of hnRNP A1 in the transportation between the nucleus and cytoplasm [87,88]. Furthermore, certain limitations persist, as the precise role of the effector protein precipitated by LOC4191, as well as the mechanism underlying protein recognition by LOC4191, remains ambiguous.

## 5. Conclusions

Global m^6^A modification was upregulated when MAC-T cell apoptosis was induced by *E. coli*. The m^6^A modification level of LOC4191 is regulated by ALKBH5. LOC4191 with hypermethylation was recognized by hnRNP A1, inhibiting apoptosis via the Caspase 3/PARP pathway. In addition, ALKBH5 inhibits apoptosis via the MAPK pathway when the cells are induced by *E. coli*. (Figure 6).

## Figures and Tables

**Figure 1 cells-12-02604-f001:**
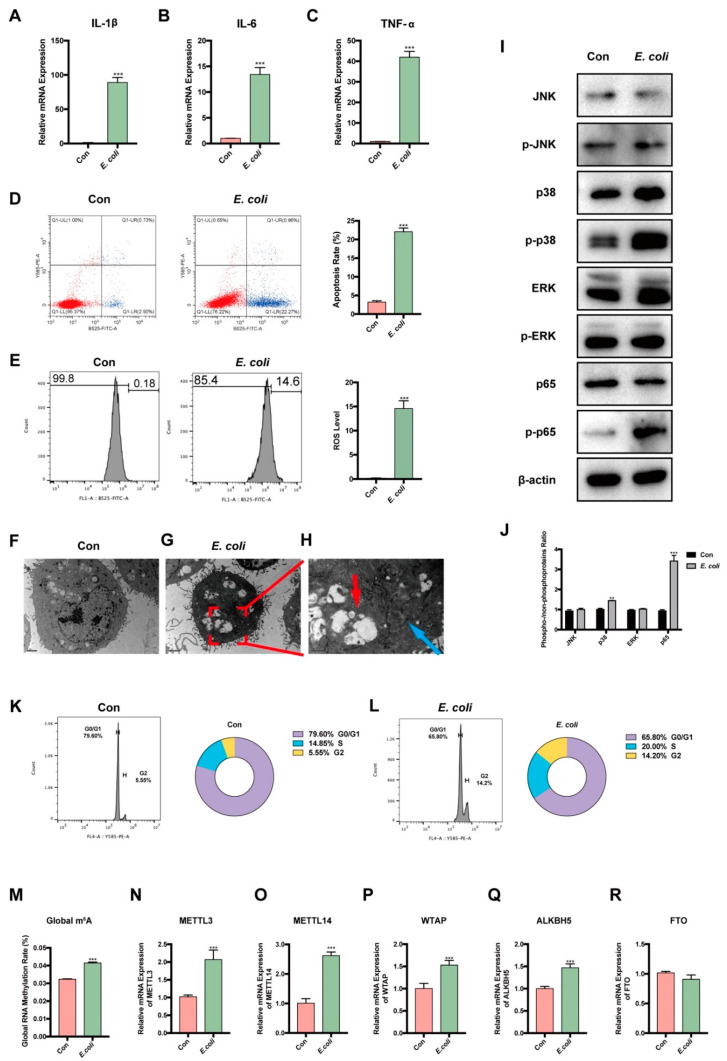
Apoptosis and involvement of m^6^A in MAC-T cells induced by *E. coli.* (**A**–**C**) IL-1β, IL-6, and TNF-α mRNA expression increased when MAC-T cells were induced by *E. coli*. (**D**) The apoptosis rate increased, and (**E**) ROS level increased in the *E. coli* group. (**F**) The TEM slides illustrated the normal MAC-T cells and (**G**) *E. coli*-induced MAC-T cells. (**H**) Vast cytosolic voids (red arrow), nuclear membrane disappearance, and chromatin condensation (blue arrow) were found. (**I**,**J**) More p38 and p65 happened to be phosphorylated, while JNK and ERK showed insignificant changes in the *E. coli* group. (**K**,**L**) The cell cycle was found to stagnate in the *E. coli* group. (**M**) Global m^6^A methylation significantly increased in the *E. coli* group. (**N**–**R**) M^6^A writer METTL3, METTL14, WTAP, and m^6^A eraser ALKBH5 increased, while FTO showed insignificant changes (** *p* < 0.01, *** *p* < 0.001).

**Figure 2 cells-12-02604-f002:**
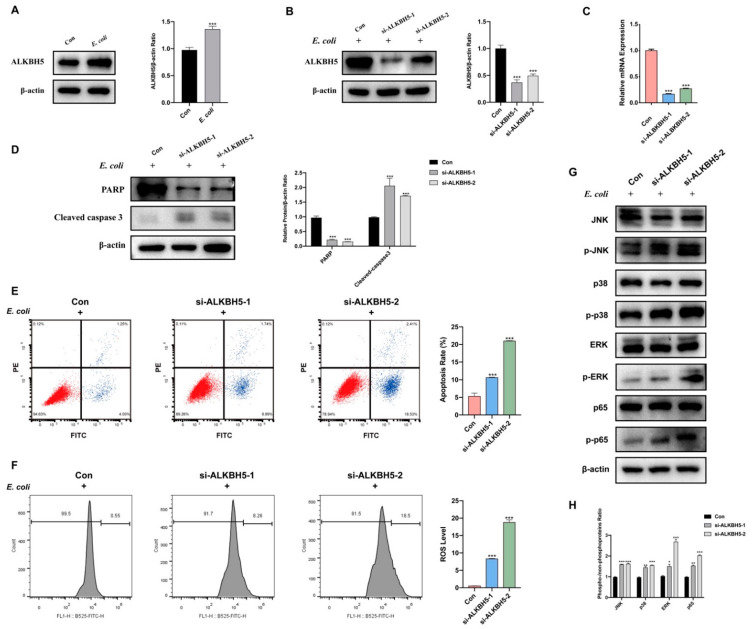
ALKBH5 silence promoted apoptosis in MAC-T cells induced by *E. coli*. (**A**) Higher ALKBH5 protein expression in the *E. coli* group. (**B**) Protein expression and (**C**) mRNA expression of ALKBH5 was successfully decreased through siRNA treatment. (**D**) Knockdown of ALKBH5 in MAC-T cells caused the expression of PARP to decrease, and cleaved caspase 3 increased when MAC-T cells were induced by *E. coli*. (**E**) Flow cytometry detected that apoptosis rate and (**F**) ROS increased in the ALKBH5 knocked-down group when MAC-T cells were induced by *E. coli*. (**G**,**H**) Western blot detected that more JNK, p38, ERK, and p65 happened to be phosphorylated in the ALKBH5 knocked-down group when MAC-T cells were induced by *E. coli* (* *p* < 0.05, ** *p* < 0.01, *** *p* < 0.001).

**Figure 3 cells-12-02604-f003:**
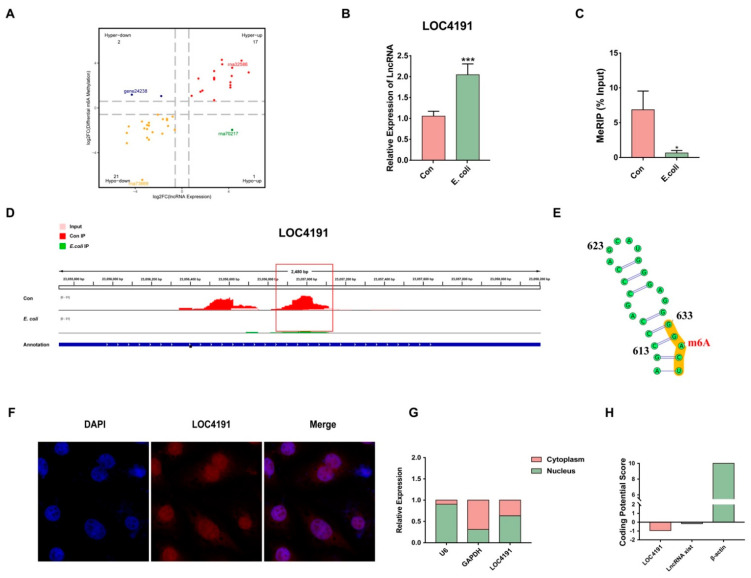
The characteristics of m^6^A-modified LOC4191 in MAC-T cells. (**A**) Conjoint analysis between the lncRNA expression level and m^6^A modification level in the *E. coli* group. (**B**) LOC4191 was upregulated in the *E. coli* group. (**C**) The m^6^A modification level of LOC4191 decreased when MAC-T cells were induced by *E. coli*. (**D**) IGV displayed the location of the m^6^A peak in LOC4191. (**E**) The specific site of m^6^A was illustrated. (**F**) FISH and (**G**) cytoplasm/nucleus isolation showed that LOC4191 closely coincided with the position of the nucleus. (**H**) CPC2 predicted that LOC4191 was disabled in the function of translation (* *p* < 0.05, *** *p* < 0.001).

**Figure 4 cells-12-02604-f004:**
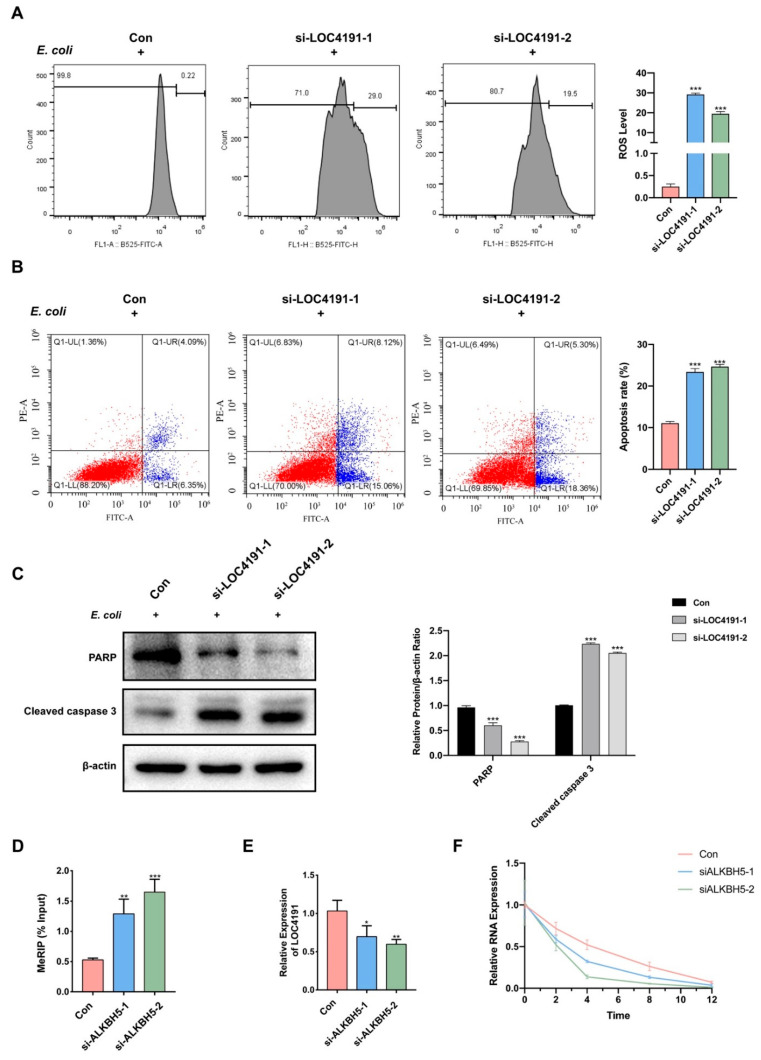
M^6^A-modified LOC4191 was the target of ALKBH5. (**A**) Flow cytometry detected that ROS and (**B**) apoptosis rates increased in the LOC4191 knocked-down group when MAC-T cells were induced by *E. coli*. (**C**) Western Blot detected that PARP was downregulated, and cleaved caspase 3 was upregulated in the LOC4191 knocked-down group when MAC-T cells were induced by *E. coli*. (**D**) Knockdown of ALKBH5 caused the m^6^A modification level in LOC4191 to increase, which led to (**E**) downregulated expression of LOC4191. (**F**) LOC4191 showed a faster rate of degeneration when ALKBH5 was knocked down (* *p* < 0.05, ** *p* < 0.01, *** *p* < 0.001).

**Figure 5 cells-12-02604-f005:**
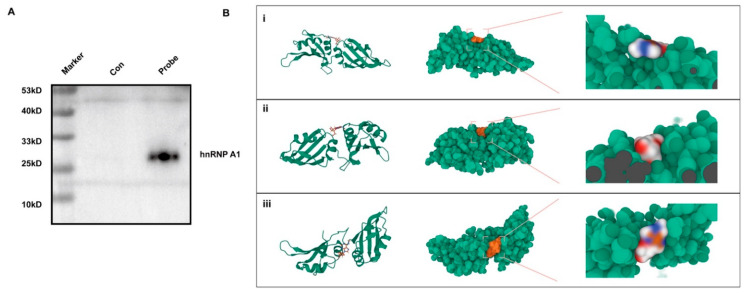
M^6^A-modified LOC4191 was recognized by the HNRNPA1 reading protein. (**A**) Western Blot was performed to verify that HNRNPA1 was present in the LOC4191-probe group. (**B**) The binding site of m^6^A in HNRNPA1 was predicted (i: Front, ii: Reverse, iii: Top). White represents the carbon atom, blue represents the hydrogen atom, and red represents the oxygen atom.

**Figure 6 cells-12-02604-f006:**
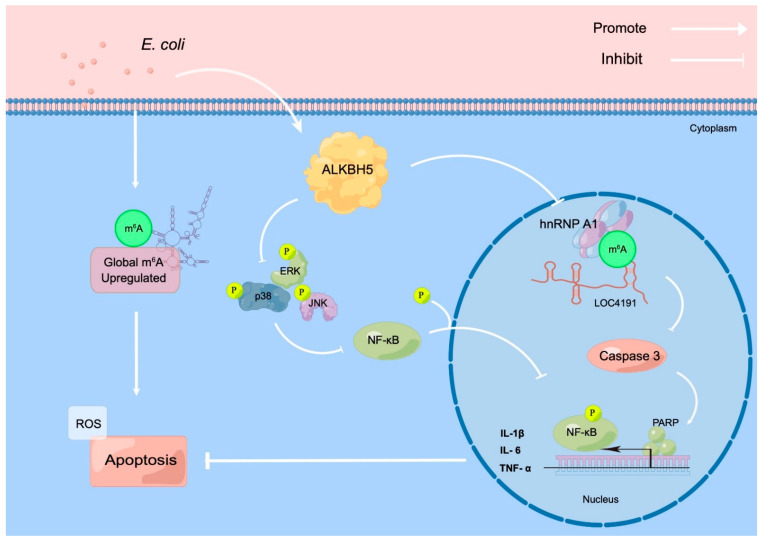
ALKBH5 stabilizes m^6^A-modified LOC4191 to suppress *E. coli*-induced apoptosis. ALKBH5 inhibits the phosphorylation of NF-κB through the MAPK pathway during the process. The m^6^A modification level of LOC4191 is regulated by ALKBH5. LOC4191 with hypermethylation inhibits apoptosis via the Caspase 3/PARP pathway, and the process was recognized by the hnRNP A1 reading protein.

**Table 1 cells-12-02604-t001:** Top 10 proteins pulled down by LOC4191.

Num	Protein Name	GO Function (Biological Process)	KEGG Function
1	Heat shock protein HSP 90-beta	Immune system process; biological process involved in interspecies interaction between organisms; biological regulation; reproductive process; cellular process; biological adhesion; response to stimulus; developmental process	Prostate cancer; Fluid shear stress and atherosclerosis; Chemical carcinogenesis-receptor activation; Antigen processing and presentation; Protein processing in endoplasmic reticulum; Progesterone-mediated oocyte maturation; IL-17 signaling pathway; Pathways in cancer; Lipid and atherosclerosis; Th17 cell differentiation; Necroptosis; Estrogen signaling pathway; Salmonella infection; NOD-like receptor signaling pathway; PI3K-Akt signaling pathway
2	Actin, cytoplasmic 2	Developmental process; biological regulation; response to stimulus; cellular process; biological adhesion; rhythmic process; immune system process	Amyotrophic lateral sclerosis; Gastric acid secretion; Influenza A; Viral myocarditis; Apoptosis; Thyroid hormone signaling pathway; Neutrophil extracellular trap formation; Phagosome; Rap1 signaling pathway; Fluid shear stress and atherosclerosis; Regulation of actin cytoskeleton; Bacterial invasion of epithelial cells; Arrhythmogenic right ventricular cardiomyopathy; Leukocyte transendothelial migration; Platelet activation; Oxytocin signaling pathway; Yersinia infection; Salmonella infection; Hippo signaling pathway; Tight junction; Proteoglycans in cancer; Dilated cardiomyopathy; Adherens junction; Hepatocellular carcinoma; Focal adhesion; Hypertrophic cardiomyopathy; Thermogenesis
3	Acetyl-CoA carboxylase 1	Localization; metabolic process; response to stimulus; cellular process; biological regulation	Alcoholic liver disease; AMPK signaling pathway; Glucagon signaling pathway; Metabolic pathways; Fatty acid biosynthesis; Pyruvate metabolism; Propanoate metabolism; Insulin signaling pathway; Fatty acid metabolism
4	Pyruvate carboxylase	Biological regulation; biological process involved in interspecies interaction between organisms; cellular process; viral process; metabolic process	Biosynthesis of amino acids; Metabolic pathways; Citrate cycle (TCA cycle); Carbon metabolism; Pyruvate metabolism
5	Annexin A2	Localization; biological regulation; biomineralization; developmental process; response to stimulus; cellular process; biological adhesion	Salmonella infection
6	Heat shock cognate 71 kDa protein	Developmental process; cellular process; response to stimulus; rhythmic process; metabolic process; reproductive process; biological process involved in interspecies interaction between organisms; biological regulation; multicellular organismal process; localization	Protein processing in endoplasmic reticulum; Legionellosis; Antigen processing and presentation; MAPK signaling pathway; Longevity regulating pathway-multiple species; Spliceosome; Endocytosis; Estrogen signaling pathway; Measles; Toxoplasmosis; Lipid and atherosclerosis; Prion disease
7	Insulin-like growth factor 2 mRNA binding protein 2	Developmental process; biological regulation; localization	--
8	Tubulin beta-5 chain	Response to stimulus; cellular process; biological regulation; biological process involved in interspecies interaction between organisms; immune system process; localization; metabolic process	Amyotrophic lateral sclerosis; Prion disease; Gap junction; Alzheimer disease; Pathways of neurodegeneration-multiple diseases; Salmonella infection; Phagosome; Huntington disease; Parkinson’s disease
9	Tubulin alpha chain	Developmental process; biological regulation; response to stimulus; cellular process; localization	Parkinson disease; Huntington disease; Phagosome; Pathways of neurodegeneration-multiple diseases; Gap junction; Alzheimer disease; Prion disease; Amyotrophic lateral sclerosis; Tight junction; Apoptosis; Salmonella infection
10	Glyceraldehyde-3-phosphate dehydrogenase	Response to stimulus; cellular process; biological regulation; metabolic process	Diabetic cardiomyopathy; Metabolic pathways; Biosynthesis of amino acids; Glycolysis/Gluconeogenesis; HIF-1 signaling pathway; Alzheimer disease; Salmonella infection; Carbon metabolism

## Data Availability

Data from “Genome—wide analysis of lncRNA in bovine mammary epithelial cell injuries induced by *Escherichia coli* and *Staphylococcus aureus*” “https://www.ncbi.nlm.nih.gov/geo/query/acc.cgi?acc=GSE181464 (accessed on 6 August 2021)”. Data from “N6-methyladenosine-modified circRNA in the bovine mammary epithelial cells injured by *Staphylococcus aureus* and *Escherichia coli*” “https://www.ncbi.nlm.nih.gov/geo/query/acc.cgi?acc=GSE196736 (accessed on 23 February 2022)”.

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
