# Peer review of "ALKBH5 Stabilized N6-Methyladenosine—Modified LOC4191 to Suppress E. coli-Induced Apoptosis"

_cells, 2023, doi:10.3390/cells12222604_

Round 1

Reviewer 1 Report

Comments and Suggestions for Authors

The authors of “ALKBH5 Stabilized N6-methyladenosine-modified LOC4191 to Suppress E. coli-induced Apoptosis” investigated the correlation of an E. coli infection and the regulation of apoptosis through m6A and ALKBH5. The manuscript describes a correlation of the infection with the altered m6A levels in lncRNAs, that in turn is linked to ALKBH5 function and downstream apoptosis processes.

Although the general finding of the manuscript describes an interesting finding and a potential regulatory function of m6A within lncRNAs and affiliated processes. Many experiments have been carried out that describe the correlation between infection, M6A levels, ALKBH5 and apoptosis. However, there are some major points that need to be addressed. To me the major problem is that it is difficult to follow the experiments and conclusion as they are only very minimally described and sometimes it is hard to follow the “logic” of the experiments and conclusion. Many experiments are depicted but not well described to allow an understanding of the proposed mechanisms and correlation. This should definitely be improved.

Major points:

1.     The method section is way too short. No references are provided and also not details on the procedures. Therefore it is impossible to comprehend how the experiments were carried out. Even if kits were used, the protocols still often need to be adjusted and optimized. Therefore more information is needed! What is MOI, was the incubation carried out for 24 h or 10 h (why for what experiment), details on the Fish experiments, etc…

2.     The numbers for the m6A peaks varies in chapter 3.1 and 3.2 and in the supplemental figures (chapter 3.1.:923 m6A peaks in 282 lncRNAs; in chapter 3.2: 920 peaks in 288 lncRNAs; numbers also do not match with figure S1). Please be more detailed of why different numbers are provided and clarify.

3.     Several figures are way too small. Neither on the prints nor on the digital version the description within the figures (e.g. 1D, 1E, 2E, 2F, 3A…) can be read.

4.     In Figure 1F it is not clear what the arrows indicate. Information is not provided in the figure legends. Generally, the figure legends are too short and not very informative.

5.     The quantifications of the Western blots are not clear to me. In Fig. 2b the quantifications seems not to correlate with the blot. For me it is hard to see that the knock down by si-ALKBH1 only reduced ALKBH5 by 50%. I am not sure if the Western blots in general are somewhat overexposed/saturated. Please make sure that this is not the case.

6.     Why is the less efficient knock down of ALKBH 5 more efficient in its effect (Fig. 2G)?

7.     The number of repeats for most experiments is missing! At least I did not find it! What about the stability of the LOC4191, is this a single experiment (Figure 4F)?

8.     The docking of m6A to hnRNP A1 seems strange, as m6A would always found within an RNA molecule. How does a single deoxy m6A molecule reflect the natural situation? What can be learned from that?

9.     Figure 5A: what is the control, what is the probe group? hnRNP will most likely bind many RNAs. Were other lncRNAs or scrambled RNAs tested as well? How does the methylation level impact the binding, since a m6A dependent interaction is proposed (docking experiments).

10.  Why was hnRNPA1 chosen? It is not in the top 10 list and the authors do not provide any information why they focused on that. Why LOC4191 and not any other lncRNA which was identified?

Comments on the Quality of English Language

The quality of English is sufficient and only minor checking is required (e.g. typos).

Author Response

Dear Reviewer,

Thank you very much for your work and the comments concerning our manuscript. Those comments are valuable and very helpful. We have read through comments carefully and have made corrections. Based on the instructions provided, we uploaded the file of the revised manuscript. Revisions in the text are shown using highlight for additions, and strikethrough font for deletions. We would love to thank you for allowing us to resubmit a revised copy of the manuscript and we highly appreciate your time and consideration. The responses are as follow.

  1. The method section is way too short. No references are provided and also not details on the procedures. Therefore it is impossible to comprehend how the experiments were carried out. Even if kits were used, the protocols still often need to be adjusted and optimized. Therefore more information is needed! What is MOI, was the incubation carried out for 24 h or 10 h (why for what experiment), details on the Fish experiments, etc…

Response: Thank you for your comment. The details of the experimental methods are added as follows.

(1) In chapter 2.2 “Inactivated E. coli-induced Apoptosis in MAC-T cells”, we have added sentences “E. coli was inactivated by 63 °C.....” (lines 82-85).

(2) We have added chapter 2.3 “siRNA Transfection” and showed the sequence of the siRNA used in this study (lines 87-94).

(3) In chapter 2.4 “RNA Extraction and RT-qPCR”, we have added sentences “Cold PBS (HyClone, China) were used to........” (lines 110-119); “Reverse transcription of the RNA......” (lines 124-129); “The expression of cDNA in different group......” (lines 130-136).

(4) In chapter 2.5 “Flow Cytometry”, we have added sentences “Apoptosis detection: Cold PBS......” (lines 139-176)

(5) In chapter 2.6 “Transmission Electron Microscopy”, we have added sentences “. Dehydration was performed......” (lines 178-181).

(6) In chapter 2.7 “Western Blot”, we have added sentences “RIPA reagent (Sigma, USA) containing protease inhibitors......” (lines 185-196).

(7) In chapter 2.8 “FISH”, we have added sentences “5×104 cells were seeded in.......” (lines 201-210).

(8) In chapter 2.9 “RNA Pull Down”, we have added sentences “The interaction between LOC4191 and.......” (lines 213-260).

(9) In chapter 2.10 “Molecular Docking Analysis”, we have changed “N6-me-dA” to “N6-methyladenosine” (line 262). The Pubchem ID was added “(CID, 102157)” (line 263).

(10) In chapter 2.11 “M6A-Modified lncRNA Library Construction”, we have added sentences “A total of 6 samples......” (line 271) and “The raw sequencing data contain.......” (lines 278-282)

  1. The numbers for the m6A peaks varies in chapter 3.1 and 3.2 and in the supplemental figures (chapter 3.1.:923 m6A peaks in 282 lncRNAs; in chapter 3.2: 920 peaks in 288 lncRNAs; numbers also do not match with figure S1). Please be more detailed of why different numbers are provided and clarify.

Response: Thank you very much for pointing out our careless mistake.

(1) The m6A peaks in chapter 3.1 is verified to exist in the control group and E. coli treatment group. We have reanalyzed the MeRIP-seq data in chapter 3.1, and a new supplementary file “Table S2 Methylated RNA sites within LncRNA.xlsx” was uploaded.

The data in the chapter 3.1 has been corrected to “714 m6A peaks in 649 lncRNAs in the control group”, “345 peaks in 324 lncRNAs in the E. coli group”, and “286 peaks were identified in 276 lncRNAs in both groups” (lines 294-296), “the predominance of chromosome 3 in the control group and chromosome 10 in the E. coli group” (lines 310-311). Moreover, Figure S1 was renewed.

(2) The m6A peaks in chapter 3.2 is the significantly different peaks in E. coli group compared with control group (P-value < 0.00001, Fold change > 2). All the differential peaks are available in “Table S3 Differentially m6A methylated sites of lncRNA in MAC-T cells induced by E. coli.xlsx”.

  1. Several figures are way too small. Neither on the prints nor on the digital version the description within the figures (e.g. 1D, 1E, 2E, 2F, 3A…) can be read.

Response: Thank you for your comment. We have readjusted the size of every figure and enlarge the description within the figures (e.g. 1A, 1B, 1C, 1D, 1E, 2A...).

  1. In Figure 1F it is not clear what the arrows indicate. Information is not provided in the figure legends. Generally, the figure legends are too short and not very informative.

Response: Thank you for your comment. We have added information in the figure legend “Vast cytosolic voids (red arrow) ......” (lines 366-367). We also enlarged the arrows and change them to a more prominent color in Figure 1F.

  1. The quantifications of the Western blots are not clear to me. In Fig. 2b the quantifications seems not to correlate with the blot. For me it is hard to see that the knock down by si-ALKBH1 only reduced ALKBH5 by 50%. I am not sure if the Western blots in general are somewhat overexposed/saturated. Please make sure that this is not the case.

Response: Thank you for your comment.

(1) We have reanalyzed the Western blots image in Figure 2B. We calculated the grey value through dividing ALKBH5 by β-actin. The efficiency of si-ALKBH5-1 could reduce the expression of ALKBH5 to 35%, while si-ALKBH5-2 could reduce the expression of ALKBH5 to 50%.

(2) The quantification of the Western blot image has been corrected (Figure 2B).

  1. Why is the less efficient knock down of ALKBH 5 more efficient in its effect (Fig. 2G)?

Response: Thank you for your comment.

We have carefully checked our results and we are confirmed about the knock down efficiency of 2 siRNAs is different. The knockdown efficiency of siALKBH5-2 is less than that of siALKBH5-1, but its impact on apoptosis is stronger.

We believe that the reason is as follow. The remaining function of ALKBH5 seems not positively correlated with its amount, since siRNA treatment could not completely inhibit the expression of ALKBH5. Hu Y, et al. used 2 shRNAs to knock down ALKBH5. shALKBH5-2, which is more efficient in knocking down ALKBH5, does not promote the invasion ability of cells as well as shALKBH5-1 (Figure 2D,H) [1].

To solve this confusion, we have added the siRNA information in chapter 2.3 “siRNA Transfection” (lines 88-94). Moreover, we have discussed about this result in the discussion section, and we would like to knock out ALKBH5 to further explore the role of ALKBH5. The sentences “To further confirmed......” was added (lines 524-531).

Reference: [1] Hu Y, Gong C, Li Z, Liu J, Chen Y, Huang Y, Luo Q, Wang S, Hou Y, Yang S, Xiao Y. Demethylase ALKBH5 suppresses invasion of gastric cancer via PKMYT1 m6A modification. Mol Cancer. 2022 Feb 3;21(1):34. doi: 10.1186/s12943-022-01522-y. PMID: 35114989; PMCID: PMC8812266.

  1. The number of repeats for most experiments is missing! At least I did not find it! What about the stability of the LOC4191, is this a single experiment (Figure 4F)?

Response: Thank you for your comment.

We have carefully checked all our results. We found that Figure 4F (the stability of the LOC4191), Figure 3G and Figure 3H did not contain error bar.

Repeated experiment was conducted to test the stability of the LOC4191 again, and we corrected Figure 4F. The finding is correlated with our previous result.

In Figure 3G, we performed the experiment for 3 times and draw the figure based on the average value. We politely think that the stacked bar chart would be more informative. In Figure 3F, we analyzed the coding ability of LOC4191 on online database (http://cpc2.gao-lab.org), and we got a specific coding potential score.

  1. The docking of m6A to hnRNP A1 seems strange, as m6A would always found within an RNA molecule. How does a single deoxy m6A molecule reflect the natural situation? What can be learned from that?

Response:Thank you for your comment. A single deoxy m6A molecule is indeed not suitable in our docking analysis.

We have corrected the docking analysis. We change the “N6-me-dA” to “N6-methyladenosine” and the m6A molecular structure was obtained from PubChem Compounds (CID, 102157) (lines 262-263). Moreover, we corrected the binding energy between hnRNP A1 and m6A to -5.564 kcal/mol (line 465)

  1. Figure 5A: what is the control, what is the probe group? hnRNP will most likely bind many RNAs. Were other lncRNAs or scrambled RNAs tested as well? How does the methylation level impact the binding, since a m6A dependent interaction is proposed (docking experiments).

Response:Thank you for your comment.

(1) In Figure 5A, the control group contains the RNA-binding protein pulled down by LOC4191 antisense probe. The probe group contains the RNA-binding protein pulled down by LOC4191 sense probe.

(2) We did not try any other lncRNAs in this study. We are very willing to perform an immunoprecipitation experiment of hnRNP in our future study and screen for its target RNA. Sentences were added in the discussion section “We successfully screened a.......” (lines 563-565).

(3) According to docking analysis, hnRNPA1 was predicted to bind to m6A. In our study, we demonstrated an increase in the m6A modification level of LOC4191 after knocking down demethylase ALKBH5. This leads to a decrease in LOC4191 stability (Figure 4F). hnRNP A1 has been verified that it could decrease the stability of its target gene [1]. Therefore, we speculate that the ability of hnRNP A1 binding to LOC4191 with hypermethylation increase during this process. Sentences in the discussion section was added “In our study, we demonstrated......” (lines 577-579), “Even though we revealed that......” (lines 583-586).

Reference: [1] Kumar P P, Emechebe U, Smith R, Franklin S, Moore B, Yandell M, Lessnick SL, Moon AM. Coordinated control of senescence by lncRNA and a novel T-box3 co-repressor complex. Elife. 2014 May 29;3:e02805. doi: 10.7554/eLife.02805. PMID: 24876127; PMCID: PMC4071561.

  1. Why was hnRNPA1 chosen? It is not in the top 10 list and the authors do not provide any information why they focused on that. Why LOC4191 and not any other lncRNA which was identified?

Response: Thank you very much for your comment.

(1) To explore how m6A-modified LOC4191 could be recognized, we hope to find a specific reading protein recognizing m6A modification. As hnRNP A1 is typical reading protein in m6A modification. We would like to focus on it in our study. Through RNA pull down, molecular docking analysis and Western blot, we verified that hnRNP A1 is interacted with m6A-modified LOC4191.

Moreover, hnRNP A1 is a reading protein localized to the nucleus. This is consistent with the subcellular localization of LOC4191 in the nucleus (Figure 3F, G). Moreover, it has been proved that hnRNP A1 can regulate the stability of target genes through m6A mechanism. In our study, we also found the stability of LOC4191 with hypermethylation decreased. Based on this, we speculate that hnRNP A1 is a reading protein of m6A-modified LOC4191. We have added sentences in the discussion section “In our study, Mass spectrometry was......” (lines 553-557), “Convincingly, hnRNP A1 is a reading protein......” (lines 554-558)

(2) The reason why we chose LOC4191 is based on our previous sequencing data. We have identified 20 lncRNAs and found them showing differential expression when MAC-T cells were induced by E. coli (Figure S3). But as the experiment progressed, we found that LOC4191 had the greatest effect on E.coli-induced apoptosis. Therefore, only in-depth mechanistic studies of LOC4191 have been conducted.

Reviewer 2 Report

Comments and Suggestions for Authors

This is interesting paper indicating that there is relationship between N6 -methyladenine modified IncRNAs and E.coli. In introduction there is a need to explain that used abbreviation of ALKBH is actually a mammmalian RNA demethylase that impacts RNA metabolism. There is also no explanation that    N6-methyladenosine and m6A mean the same. It may create a problems for readers less oriented in the presented topic. Methods are well presented. Also results together with attached original images for blots are convincing. Conclusions have to be changed/ enlarged. Only one sentence with attached figure is not enough. Figure 6 has to be described in details. I recommend to accept the manuscript after minor revision.   

Author Response

Dear Reviewer,

Thank you very much for your work and the comments concerning our manuscript. Those comments are valuable and very helpful. We have read through comments carefully and have made corrections. Based on the instructions provided, we uploaded the file of the revised manuscript. Revisions in the text are shown using highlight for additions, and strikethrough font for deletions. We would love to thank you for allowing us to resubmit a revised copy of the manuscript and we highly appreciate your time and consideration. The responses are as follow.

Comment 1. In introduction there is a need to explain that used abbreviation of ALKBH is actually a mammalian RNA demethylase that impacts RNA metabolism.

Response: Thank you for your comment. We have explained that the full name of ALKBH5 is AlkB Homolog 5, and it is a mammalian RNA demethylase in the introduction section. Sentences were added “AlkB Homolog 5 (ALKBH5) is a mammalian......” (line 57). Moreover, we have added several abbreviations (lines 620-623).

Comment 2. There is also no explanation that N6-methyladenosine and m6A mean the same. It may create a problem for readers less oriented in the presented topic.

Response: Thank you for your comment. We have corrected that m6A is the abbreviation of N6-methyladenosine in the introduction section. We added sentences “N6- methyladenosine (m6A) is......” (line 53).

Comment 3. Conclusion has to be changed/ enlarged. Only one sentence with attached figure is not enough. Figure 6 has to be described in detail.

Response: Thank you for your comment. Conclusion has been added more details. We have added sentences “M6A modification level of LOC4191......” (lines 594-597). Moreover, the legend of Figure 6 has been added more information “Figure 6. ALKBH5 stabilizes m6A-modifie.......” (lines 613-617)

Reviewer 3 Report

Comments and Suggestions for Authors

The present manuscript "ALKBH5 Stabilized N6-methyladenosine-modified LOC4191 to 2 Suppress E. coli-induced Apoptosis" describes the mechanism of m6A on lncRNA in cells under E. coli-induced apoptosis. The aim of the study is very interesting but I have some concerns:

1) Why did the author select E.coli as bacterial model? What about other bacterial models? The rationale is missing.

2) The introduction is missing of infectious diseases background on m6A. Some recent studies are totally absent:

  • DOI: 10.3390/ph14111135; 10.1007/s11010-023-04841-w; 10.1016/j.virol.2023.03.007

3) Materials and methods section is very poorly described. Please improve it.

Author Response

Dear Reviewer,

Thank you very much for your work and the comments concerning our manuscript. Those comments are valuable and very helpful. We have read through comments carefully and have made corrections. Based on the instructions provided, we uploaded the file of the revised manuscript. Revisions in the text are shown using highlight for additions, and strikethrough font for deletions. We would love to thank you for allowing us to resubmit a revised copy of the

manuscript and we highly appreciate your time and consideration. The responses are as follow.

1) Why did the author select E.coli as bacterial model? What about other bacterial models? The rationale is missing.

Response:Thank you for your comment.

(1) Although several other pathogens were also isolated from bovines with mastitis, including Streptococcus and Staphylococcus aureus, these bacteria usually only cause mild mastitis.

(2) E. coli has been identified as the prevalent pathogen responsible for mastitis in dairy cows, and the contaminated milk caused by E. coli poses a significant threat to human health. Since over 50% of severe mastitis is caused by E. coli, we choose E. coli rather than other bacteria as bacterial model for further exploring. Furthermore, we have added sentences in the introduction section “Although several other pathogens were also isolated......” (lines 40-44). Moreover, E. coli is a typical Gram-negative bacterium, and it is the most widely studied model bacteria. The cell membrane of E. coli contains antigens such as LPS that induce serious inflammation, while other Gram-positive bacteria mainly secrete exotoxins. Sentences in the discussion section were added “In pathogenesis, E. coli is a typical......” (lines 487-490). Overall, E. coli is a good bacterial model.

2) The introduction is missing of infectious diseases background on m6A. Some recent studies are totally absent:

  • DOI: 10.3390/ph14111135; 10.1007/s11010-023-04841-w; 10.1016/j.virol.2023.03.007

Response: Thank you for your comment. We have added the infectious diseases background on m6A in the introduction section. We added the sentences “Plenty of research have shown...... [2-4]” (lines 55-56).

Additional Reference (lines 854-859):

  1. Ge, Y.; Tang, S.; Xia, T.; Shi, C. Research progress on the role of RNA N6-methyladenosine methylation in HCV infection. Virology 2023, 582, 35-42, doi:10.1016/j.virol.2023.03.007.
  2. Zannella, C.; Rinaldi, L.; Boccia, G.; Chianese, A.; Sasso, F.C.; De Caro, F.; Franci, G.; Galdiero, M. Regulation of m6A methylation as a new therapeutic option against COVID-19. Pharmaceuticals (Basel) 2021, 14, doi:10.3390/ph14111135.
  3. Zhu, L.; Zhang, H.; Zhang, X.; Xia, L. RNA m6A methylation regulators in sepsis. Mol Cell Biochem 2023, 582, 35-24, doi:10.1007/s11010-023-04841-w.

3) Materials and methods section is very poorly described. Please improve it.

Response: Thank you for your comment. The details of the experimental methods are added as follow.

(1) In chapter 2.2 “Inactivated E. coli-induced Apoptosis in MAC-T cells”, we have added sentences “E. coli was inactivated by 63 °C.....” (lines 82-85).

(2) We have added chapter 2.3 “siRNA Transfection” and showed the sequence of the siRNA used in this study (lines 87-94).

(3) In chapter 2.4 “RNA Extraction and RT-qPCR”, we have added sentences “Cold PBS (HyClone, China) were used to........” (lines 110-119); “Reverse transcription of the RNA......” (lines 124-129); “The expression of cDNA in different group......” (lines 130-136).

(4) In chapter 2.5 “Flow Cytometry”, we have added sentences “Apoptosis detection: Cold PBS......” (lines 139-176)

(5) In chapter 2.6 “Transmission Electron Microscopy”, we have added sentences “. Dehydration was performed......” (lines 178-181).

(6) In chapter 2.7 “Western Blot”, we have added sentences “RIPA reagent (Sigma, USA) containing protease inhibitors......” (lines 185-196).

(7) In chapter 2.8 “FISH”, we have added sentences “5×104 cells were seeded in.......” (lines 201-210).

(8) In chapter 2.9 “RNA Pull Down”, we have added sentences “The interaction between LOC4191 and.......” (lines 213-260).

(9) In chapter 2.10 “Molecular Docking Analysis”, we have changed “N6-me-dA” to “N6-methyladenosine” (line 262). The Pubchem ID was added “(CID, 102157)” (line 263).

(10) In chapter 2.11 “M6A-Modified lncRNA Library Construction”, we have added sentences “A total of 6 samples......” (line 271) and “The raw sequencing data contain.......” (lines 278-282)

Round 2

Reviewer 1 Report

Comments and Suggestions for Authors

The authors largely addressed the points that were raised. I still believe that the size of the fonds in some figures are too small. In addition, I still do not see the point of simulating a single m6A without the context of the RNA strand, which seems quite artificial. However, this is only one part of the indications in a line of experiments.

Overall, the manuscript improved considerably!

All the best!

Reviewer 3 Report

Comments and Suggestions for Authors

The manuscript has been deeply improved and now is ready for publication.